# Tumor Suppressive Effects of GAS5 in Cancer Cells

**DOI:** 10.3390/ncrna8030039

**Published:** 2022-05-28

**Authors:** Jesminder Kaur, Nur’ain Salehen, Anwar Norazit, Amirah Abdul Rahman, Nor Azian Abdul Murad, Nik Mohd Afizan Nik Abd. Rahman, Kamariah Ibrahim

**Affiliations:** 1Department of Biomedical Science, Faculty of Medicine, Universiti Malaya, Kuala Lumpur 50603, Malaysia; jesminkaur18@gmail.com (J.K.); nurain_36@um.edu.my (N.S.); anwar.norazit@um.edu.my (A.N.); 2Department of Biochemistry and Molecular Medicine, Faculty of Medicine, Universiti Teknologi MARA, Cawangan Selagor, Kampus Sungai Buloh, Sungai Buloh 47000, Malaysia; amirahar@uitm.edu.my; 3UKM Medical Molecular Biology Institute, National University of Malaysia, Kuala Lumpur 56000, Malaysia; nor_azian@ppukm.ukm.edu.my; 4Department of Cell and Molecular Biology, Faculty of Biotechnology and Biomolecular Sciences, Universiti Putra Malaysia, Serdang 43400, Malaysia; m.afizan@upm.edu.my

**Keywords:** lncRNA, growth-arrest specific transcript 5, GAS5, tumor suppressor, polymorphism

## Abstract

In recent years, long non-coding RNAs (lncRNAs) have been shown to play important regulatory roles in cellular processes. Growth arrests specific transcript 5 (GAS5) is a lncRNA that is highly expressed during the cell cycle arrest phase but is downregulated in actively growing cells. Growth arrests specific transcript 5 was discovered to be downregulated in several cancers, primarily solid tumors, and it is known as a tumor suppressor gene that regulates cell proliferation, invasion, migration, and apoptosis via multiple molecular mechanisms. Furthermore, GAS5 polymorphism was found to affect GAS5 expression and functionality in a cell-specific manner. This review article focuses on GAS5’s tumor-suppressive effects in regulating oncogenic signaling pathways, cell cycle, apoptosis, tumor-associated genes, and treatment-resistant cells. We also discussed genetic polymorphisms of GAS5 and their association with cancer susceptibility.

## 1. Introduction

The Encyclopaedia of DNA Elements (ENCODE) project revealed that the non- coding section of DNA, which was long regarded to be “junk DNA,” is now recognized as functional elements in the human genome and produces functional non-coding RNAs that regulate cellular functions in a variety of ways [1,2]. LncRNAs are non-coding RNAs that contain more than 200 nucleotides and are the result of extensive genome transcription. Even though lncRNAs are not translated into functional proteins, they are referred to as regulatory RNAs because of their role in gene regulation via epigenetic, transcription, and post-transcriptional activities [3]. Recent studies have indicated that the dysregulation of lncRNA expression can result in tumor initiation and progression [4]. LncRNA GAS5 is located at chromosome 1q25 [5] and belongs to the 5’-terminal oligopyrimidine (5’TOP) gene family, which encodes 10 box C/D short nucleolar RNAs (snoRNAs) within the introns (Figure 1) [6]. GAS5 was initially isolated and discovered from a group of genes expressed only during the growth arrest phase of NIH 3T3 cells [7]. During the growth-arrested phase, spliced GAS5 mRNA accumulates into messenger ribonucleoprotein (mRNP) particles, sequestered away from active ribosomes and remains untranslated [6], resulting in higher GAS5 expression in growth-arrested cells than in actively growing cells. However, in growth-induced cells, transcriptional activity is unaffected, implying that post-transcriptional mechanisms regulate GAS5 mRNA levels [8,9]. As a member of 5′TOP mRNAs, GAS5 translation is regulated by the mammalian targets of rapamycin (mTOR) pathway [10], and its expression is influenced by the nonsense-mediated mRNA decay (NMD) pathway during active translation [11]. The NMD pathway can change GAS5 functionalities such as apoptotic-related gene transcriptional activity by regulating GAS5 expression levels and decay rates [12].

Aberrant GAS5 expression was reported in actively proliferating cancer cells including breast cancer [14], cervical cancer [15], colorectal cancer [16], gastric cancer [17], hepatocellular carcinoma [18], renal cell carcinoma [19], and glioma [20]. The prognostic value of GAS5 in cancer was explored by evaluating the relationship between the GAS5 expression with clinicopathological features and patients’ survival. Cao and colleagues [15] demonstrated that decreased GAS5 expression was negatively correlated with the FIGO (International Federation of Gynecology and Obstetrics) stage, vascular invasion, and lymph node metastasis in cervical cancer. Furthermore, studies in clear cell renal carcinoma [21] and laryngeal squamous carcinoma [22] have linked lower GAS5 expression to the advanced tumor stage and tumor size, respectively. Another study by Sun et al. showed that downregulated GAS5 was significantly correlated with the shorter survival time in patients with gastric cancer [17]. Besides, GAS5 is also known as a tumor suppressor gene due to its role in inhibiting cell proliferation, migration, invasion, and promoting apoptosis in cancer cells through binding and downregulating oncomirs [16,21,22,23]. GAS5 genetic polymorphisms, on the other hand, have recently been linked to cancer susceptibility and progression [24,25,26,27]. This review summarises GAS5’s tumor-suppressive effects in cancer cells, including modulation of oncogenic signaling pathways, cell cycle, tumor-suppressor genes, apoptosis, and treatment-resistant cells. We also discussed the impact of GAS5 polymorphisms on cancer patients. Figure 2 depicts the GAS5 secondary structure and sponging regions of microRNAs (miRNAs).

## 2. GAS5 Expression Is Regulated by the mTOR Signaling Pathway

The 70-kDa ribosomal protein S6 kinase (p70S6K) and eukaryotic initiation factor 4E/4E-binding protein 1 (eIF4E/4E-BP1) are two independent mTOR downstream effectors that are involved in cell cycle control [29]. In response to mitogenic stimulation or nutrient availability, mTOR downstream signaling effectors regulate cell growth and cell cycle progression through controlling cellular translation [29]. The p70S6 kinase plays an important role in regulating translational of 5′TOP mRNAs through phosphorylation of ribosomal S6 protein [30]. The mTOR signaling pathway selectively regulates GAS5 translation [10] via mitogen-induced translation by p70S6K [31]. However, due to a short reading frame [32], GAS5 is subjected to rapid degradation by the NMD pathway, thus lowering GAS5 expression during the normal cell growth process [11,12]. Inhibition of GAS5 translation by the immunosuppressant rapamycin, on the other hand, can cause cell cycle arrest by increasing GAS5 levels [10].

## 3. Role of GAS5 in Oncogenic Signaling Pathways

### 3.1. PI3K/AKT/mTOR Pathway

In the activated phosphatidylinositol-3-kinase (PI3K)/AKT pathway, 4E-BP1 is phosphorylated [33] and dissociated from the eIF4E/4E-BP1 complex [34] resulting in increased cyclin D1 protein levels in response to increased eIF4E expression [35] and thus promoting the cell cycle progression [36]. GAS5 plays an important regulatory role in modulating the PI3K/AKT/mTOR pathway via sponging oncomirs. Xue et al. revealed that GAS5 overexpression in prostate cancer cells can significantly reduce the phosphorylation of AKT and its downstream proteins mTOR and S6K1 through targeting miR-103 [23]. Dong et al. have further demonstrated that the ectopic expression of GAS5 could affect gastric cancer progression through the inactivation of the AKT/mTOR pathway by negatively regulating miR-106a-5p expression [37]. Similarly, J. Liu et al. have also confirmed that GAS5 can also inactivate the PI3K/AKT pathway and inhibit cell growth in osteosarcoma cells by upregulating phosphatase and tensin homologue (PTEN) via sponging miR-23a-3p [38]. Figure 3 depicts the downregulation of GAS5 in actively growing cells.

### 3.2. PTEN/AKT Pathway

*PTEN* gene plays a critical role in controlling cell proliferation and cell growth by regulating the AKT signaling pathway [39]. The ability of PTEN to induce cell-cycle inhibition depends on the negative regulation of the PI3K/AKT signaling pathway [40]. Immunohistochemical analysis revealed that the loss of PTEN triggers AKT phosphorylation in endometrial carcinoma [41]. Meanwhile, a study in non-small cell lung cancer cells found that overexpression of PTEN inhibits cell growth by inducing G0/G1 arrest [42]. According to recent research, GAS5 may regulate PTEN expression by competitively binding to miRNAs and thus reducing PTEN suppression. Moreover, GAS5 can also sensitize the lung cancer cells to chemotherapeutic [43] and radiosensitivity treatments [44] through miR-21/PTEN/AKT axis by sponging miR-21, thereby upregulating PTEN expression and consequently suppressing AKT phosphorylation. Likewise, downregulated GAS5 in hepatocellular cancer (HCC) cells resulted in decreased PTEN levels and increased doxorubicin resistance in HCC [45]. As shown in Table 1, GAS5 inhibits tumors via sponging oncomirs and upregulating PTEN expression.

## 4. GAS5 as Part of Cell Cycle Regulatory Mechanism

### 4.1. c-Myc Expression

c-Myc is required for the activation of Cyclin-Dependent Kinase (CDK) complexes during cell growth and is selectively translated via eIF4E activation [50]. By increasing cyclin E/CDK2 function and decreasing cyclin E-associated p27 levels, c-Myc could suppress p27-induced growth arrest [51]. GAS5 was found to regulate c-Myc protein levels by interacting with eIF4E during the translation initiation phase and suppressing c-Myc translation by preventing c-Myc mRNA from entering the polysome [52]. By blocking c-Myc translation, GAS5 could potentially induce cell growth arrest.

### 4.2. CDK Inhibitors

Inhibitors such as p27kip1, p21, and cyclin-dependent kinase inhibitor 1C (CDKN1C) (known as p57kip2) are of cyclin-dependent kinases (CDK) and are negative regulators of cell cycle progression [53,54]. GAS5 induces cell growth arrest by regulating the expression of CDK inhibitors. For instance, GAS5 promotes cell growth arrest by increasing the promoter activity of p27kip1, through enhancing the binding of E2F1 to p27kip1 promoter and subsequently upregulating p27 kip1 expression [55]. Additionally, GAS5 can upregulate p27 expression by acting as a molecular sponge to miR-222-3p and abrogates its ability to inhibit p27 protein expression [56]. According to a study on stomach cancer, GAS5 modulates p21 expression and hence maintains G1 phase cell cycle arrest. Depletion of GAS5 may increase the turnover of the transcriptional activator Y-box-binding protein 1 (YBX1), reducing p21 expression and so preventing G1 phase arrest [57].

The enhancer of zeste homolog 2 (EZH2) epigenetic modifier was found to be overexpressed in melanoma, resulting in tumor growth and spread by inhibiting tumor-suppressive genes [58]. Xu et al. showed that GAS5 could reverse the effect of EZH2 in melanoma cells by recruiting transcription factor E2F4, which acts as transcriptional repressor to the EZH2 promoter region, therefore affecting the EZH2 expression from the transcriptional level and eventually leads to upregulation of tumor suppressor gene *CDKN1C* [59]. Likewise, a study in bladder cancer has further verified the GAS5 role in promoting apoptosis through inhibiting EZH2 transcription similarly by recruiting transcription factor E2F4 to EZH2 promoter [60]. Figure 4 illustrates the role of GAS5 as a guide.

## 5. GAS5 Regulates Cellular Apoptosis

Glucocorticoid receptor (GR) is a member of the nuclear receptor family and activation of GR by glucocorticoids (GCs) results in translocation of GR from the cytoplasm to the nucleus, which then interacts with glucocorticoid response elements (GREs), thereby regulating the expression of target genes [61]. In lymphoid cancer, GCs are used as anticancer drugs due to their inhibitory effects in lymphoid tissue [62]. Despite GCs’ antitumor effects in hematopoietic malignancies, a recent meta-analysis study on the significance of GR expression in cancer found that high GR expression was linked to cancer progression and a worse prognosis in certain solid tumors, including endometrial, ovarian, and early stages of untreated triple-negative breast cancers [63]. Glucocorticoids have cell-specific pro-apoptotic and anti-apoptotic effects, with lymphoid tissue showing pro-apoptotic and anti-proliferative effects and solid cancer cells showing anti-apoptotic and proliferation-promoting effects [64]. GAS5 serves as a riborepressor of the GR, suppressing GR-induced transcriptional activity by binding to the DNA binding domain of GR via its double-stranded GRE mimic sequence [65]. A stem-loop structure encoded in GAS5 exon 12 which serves as GRE-mimic was found to induce apoptosis in breast cancer cells [66]. The role of GAS5 as an apoptosis regulator was delineated by overexpressing GAS5 in breast epithelial cells which resulted in increased apoptosis in GAS5-overexpressed cells [14]. Similar findings in prostate cancer cells showed that high GAS5 levels increase both basal and drug-induced apoptosis, whereas downregulation of GAS5 attenuates apoptosis [67]. The anti-apoptotic activity of GCs was previously found to be attributable to the overexpression of the cellular inhibitor apoptosis 2 (cIAP2) [68]. Growth arrests specific transcript 5 can reduce the transcriptional activity of anti-apoptotic genes like *cIAP2* and serum- and glucocorticoid-regulated kinase 1 (*SGK1*), as well as increase cellular death [12]. Figure 5 shows how GAS5 acts as a riborepressor for GR.

## 6. Genetic Variants Affect GAS5 Expression and Cancer Susceptibility

In leukemia, GAS5 behaves differently than in solid tumors. According to Yan and colleagues, patients with GAS5 rs55829688 CC genotype exhibited higher GAS5 expression and were associated with poor prognosis in acute myeloid leukemia. Thus, overexpression of GAS5 was predicted to decrease GC function and aggravate chemotherapy-induced hematological damage [69]. On the other hand, Wang et al. demonstrated that rs55829688 CT/TT genotypes were linked to higher GAS5 expression and a greater risk of colorectal cancer when compared to the CC genotype. They also found that the rs55829688 T>C polymorphism downregulates GAS5 expression and ultimately slows CRC growth [24].

GAS5 expression was shown to be inconsistent in hepatocellular carcinoma. In previous studies, downregulation of GAS5 was found to promote HCC cell proliferation and growth [18,45]. On the other hand, Tao et al. reported that the GAS5 rs145204276 deletion allele was linked to higher GAS5 expression and an increased risk of HCC, implying that GAS5 may operate as a proto-oncogene in HCC [70]. Similarly, in oral cancer, GAS5 single nucleotide polymorphism (SNP) rs145204276 variants (Ins/Del or Del/Del) were associated with poor differentiation cell status in male patients. According to data from the GTEX database, individuals with these variants had significantly higher GAS5 expression in esophageal mucosa tissues [25]. Weng et al. revealed that cervical patients with allele deletion in the GAS5 rs145204276 are likely to have a poorer hazard ratio of 5 years survival [71]. In contrast, GAS5 rs145204276 allele deletion in cervical squamous cell carcinoma [26], urothelial cell carcinoma [24], and glioma [72] was associated with lower GAS5 expression and increased tumor risk. Likewise, GAS5 rs145204276 deletion allele was associated with a lower risk of breast cancer [73], and a lower risk of lymph node metastasis in prostate cancer [74].

In HCC [70], oral cancer [25], and uterine cervical cancer [71], the promoter variant rs145204276 may impact GAS5’s tumor suppressive effect; nevertheless, its tumor suppressive activity has been described in osteosarcoma, where allele deletion was associated with enhanced GAS5 expression and smaller tumors [27]. Apart from rs145204276, two other SNPs located at the promoter region, rs2067079 with a T allele and rs6790 with a G allele have been associated with an increased risk of bladder cancer [75]. Rakhshan et al. hypothesized that minor alleles of these SNPs could work together to disrupt GAS5 function [75]. Both SNPs have been identified as possible biomarkers for chemoradiotherapy-induced adverse responses in nasopharyngeal cancer [76]. Guo et al. have demonstrated that both SNPs may influence the transcription activity of GAS5, due to a strong expression of Quantitative Trait Locus (e-QTL) property in a variety of tissues. Furthermore, their structural analysis of GAS5 rs2067079 showed that GAS5 polymorphism has a clear effect on GAS5 secondary structure and could potentially disrupt the GAS5-miRNA sponge role through changes in the miRNA binding site [76]. We summarize that GAS5 variations may affect GAS5 function as a tumor suppressor by retaining or disrupting its function, regulating GAS5 expression, or modifying GAS5 secondary structure. The genetic variations of GAS5 in various malignancies are summarised in Table 2.

## 7. GAS5 Regulates Target Genes via Competing Endogenous RNA (ceRNA) Network

MiRNAs are the small non-coding RNAs involved in various gene regulations through repressing the translation of the mRNAs. The interplay between coding and non-coding RNAs is known as ceRNA and the regulatory crosstalk between transcriptomes is mediated by miRNAs via miRNA response elements [78]. Perturbations in ceRNA regulatory components such as lncRNAs, miRNAs, and mRNAs are associated with the pathophysiology of cancer and several other diseases [79]. Growth arrests specific transcript 5 acts as a molecular sponge in the ceRNA network, absorbing oncogenic miRNAs and preventing the repression of tumor suppressor genes involved in inhibiting cell proliferation, migration, invasion, and promoting cell apoptosis in cancers such as colorectal cancer [80], glioma stem cells [81], triple-negative breast cancer [82], and clear cell renal cell carcinoma [21]. Figure 6 depicts GAS5’s function as a molecular sponge.

### 7.1. FOXO

The forkhead box O (FOXO) is a subgroup of the forkhead-box (FOX) transcription factors superfamily [84]. FOXOs are considered tumor suppressors due to their inhibitory role in cancer. Evidence suggested that downregulation of FOXOs expression by miRNAs has been shown to facilitate cell proliferation, invasion, and metastasis in cancer cells [85]. Recent studies suggested that GAS5 could potentially upregulate FOXOs expression by abrogating the repressive function of miRNAs. A study in colorectal cancer showed that GAS5 can inhibit cell proliferation and promotes apoptosis by upregulating FOXO3a expression via sponging miR-182-5p [80]. Another study in glioma stem cells showed that GAS5 can also upregulate the other member of the FOXO family, namely FOXO1 via downregulating miR-196a-5p expression resulting in inhibition of cell proliferation, migration, invasion, and promotes apoptosis in glioma stem cells [81]. A similar study has further verified the tumor-suppressive effect of the GAS5/miR-196a-5p/FOXO1 signaling axis in triple-negative breast cancer cells [82].

### 7.2. hZIP1

Human ZIP1 (hZIP1) is known as a zinc transporter in human cells and is responsible for zinc uptake in cells [86]. The zinc status was evaluated in several cancers using in situ staining method and zinc levels were found markedly decreased in malignant cells namely hepatocellular cancer, pancreatic ductal adenocarcinoma, and breast invasive ductal adenocarcinoma as compared to their respective normal cells [87]. A study in prostate cancer indicated that the downregulation of hZIP1 transporter protein is accompanied by cellular zinc depletion in malignant prostate tissue [88]. Dong et al. reported that the hZIP1 expression was downregulated in clear cell renal cell carcinoma as compared to normal kidney samples [89]. In a separate study, Dong et al. revealed that GAS5 overexpression can upregulate hZIP1 protein expression and reduce tumorigenicity of clear cell renal cell carcinoma cells by sponging miR-223 [21].

## 8. GAS5 Modulates Chemosensitivity and Radiosensitivity in Cancer Cells

Chemotherapy and radiotherapy are used as standard treatments in cancer. However, the efficacy of these treatments is reduced in cancer resistance cells. Recently, non-coding RNAs have come to light as potential therapeutic targets for targeted and drug-resistance therapy in cancers [90,91]. This is typically due to the dysregulated non-coding RNAs that were found to have an essential role in cellular resistance towards chemotherapeutic drugs [92]. Growth arrests specific transcript 5 expression was verified to be significantly downregulated in resistant cancer cells. Its ectopic expression has been shown to improve treatment efficacy in radio- and chemo-resistant cancer cells by sponging oncomirs, thereby regulating tumor-suppressive effects by upregulating genes like reversion inducing cysteine rich protein with kazal motifs (*RECK*), dickkopf WNT signaling pathway inhibitor 2 (*DKK2*), immediate early response 3 (*IER3*) and phospholysine phosphohistidine inorganic pyrophosphate phosphatase (*LHPP*). Moreover, GAS5 can also sensitize cells to overcome treatment resistance allowing cells to undergo apoptosis [93,94,95,96]. Furthermore, via sponging miR-221 and elevating suppressor of cytokine signaling 3 (*SOCS3*) gene expression, overexpression of GAS5 was reported to inhibit gemcitabine resistance-mediated stem cell-like features, tumor metastasis, and epithelial-mesenchymal transition (EMT) of pancreatic cancer cells [97]. The roles of GAS5 in chemosensitivity and radiosensitivity regulation in cancer cells are summarised in Table 3.

## 9. Conclusions

GAS5 is downregulated in a variety of solid tumors. Recent studies showed that ectopic expression of GAS5 inhibits cell growth and enhances cell death through altering oncomirs, tumor suppressor genes, and oncogenic signaling pathways. Although GAS5 has a tumor-suppressive role in a range of solid tumors, the precise molecular mechanism by which GAS5 acts in human cancers remains unknown, especially in hematological and solid cancers with GAS5 polymorphisms. GAS5 has been discovered to perform an adversarial role in leukemia carcinogenesis control. Besides, GAS5 polymorphism has been shown to have an opposing effect in cancers such as hepatocellular carcinoma, colorectal cancer, and oral cancer, where higher GAS5 expression has been linked to an increased risk of cancer. However, the underlying mechanisms of this opposing event remain unknown, necessitating further investigation. Research in hematopoietic malignancies is still lacking and further investigation is warranted to verify GAS5 functions in hematopoietic malignancies. Based on current research, GAS5 appears to be a potential therapeutic target, particularly in terms of restoring its tumor suppressor activity to restore dysregulated signaling pathways found in many cancers.

## Figures and Tables

**Figure 1 ncrna-08-00039-f001:**
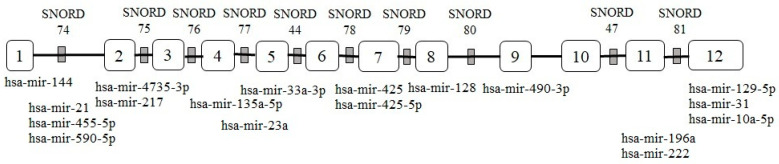
Diagram representation of GAS5 transcript (GenBank ID: NR_002578.3) with the sequence length of 656 nt encodes 10 snoRNAs within introns. The sponging regions of miRNAs predicted by miRcode [13] are located at exon 1 (hsa-mir-144), intron 1 (hsa-mir-21, hsa-mir-455-5p and hsa-mir-590-5p), exon 2 (hsa-mir-4735-3p and hsa-mir-217), exon 4 (hsa-mir-135a-5p), exon 5 (hsa-mir-33a-3p), intron 4 (hsa-mir-23a), exon 7 (hsa-mir-425 and hsa-mir-425-5p, exon 8 (hsa-mir-128), exon 9 (hsa-mir-490-3p), intron 11 (hsa-mir-196a and hsa-mir-222) and exon 12 (hsa-mir-129-5p, hsa-mir-31 and hsa-mir-10a-5p).

**Figure 2 ncrna-08-00039-f002:**
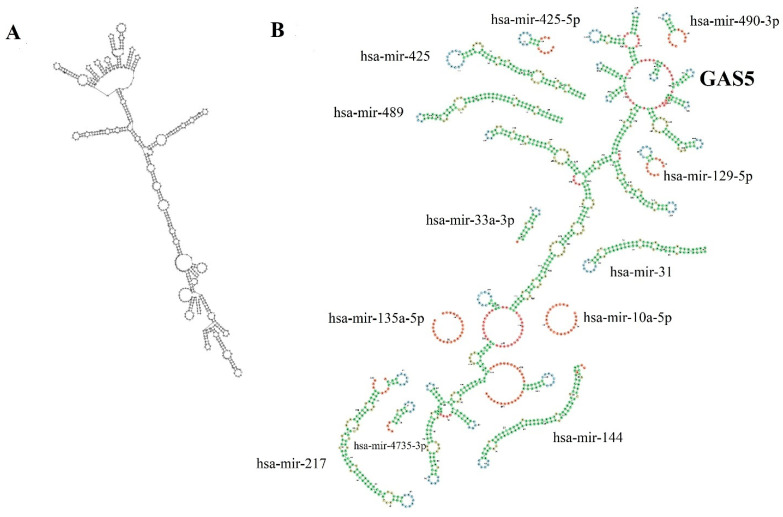
Diagram representation of (**A**) The minimum free energy (MFE) secondary structure of GAS5 predicted using RNA-fold web server (http://rna.tbi.univie.ac.at/cgi-bin/RNAWebSuite/RNAfold.cgi) (accessed on 24 March 2022). (**B**) The secondary structure of GAS5 regions sponging miRNAs in GAS5 variant 1 is shown using forna (force directed graph layout) [28].

**Figure 3 ncrna-08-00039-f003:**
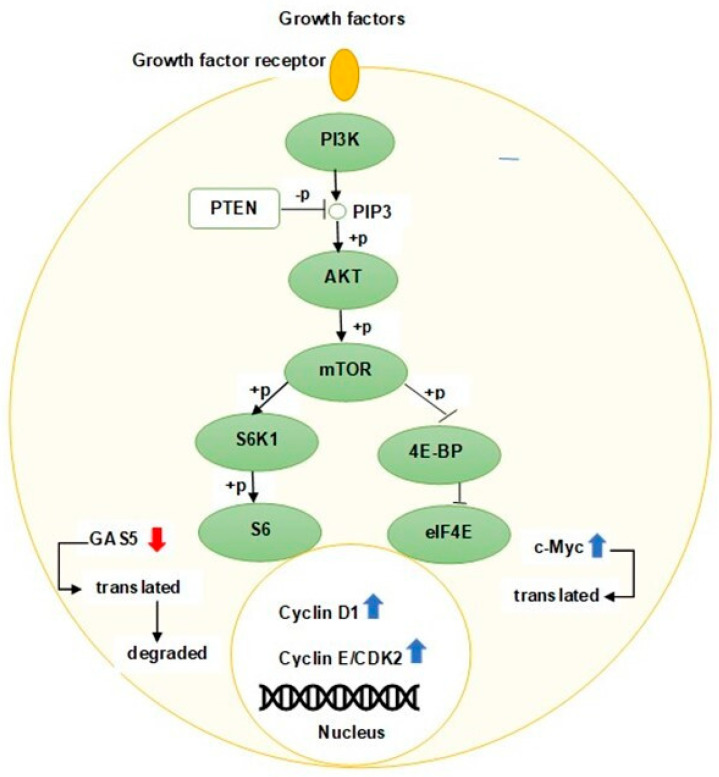
Schematic representation of proliferating cell with activated PI3K/AKT/mTOR pathway. The ribosomal S6 protein is phosphorylated (+p) when the PI3K/AKT/mTOR pathway is activated, resulting in translation and degradation of GAS5 by the NMD pathway. Downregulation of GAS5 causes an increase in c-Myc protein, thereby facilitating cell cycle progression via inducing activation of cyclin-CDK complexes. PTEN, a tumor suppressor protein, blocks the PI3K/AKT/mTOR signaling pathway by dephosphorylating (−p) PIP3.

**Figure 4 ncrna-08-00039-f004:**
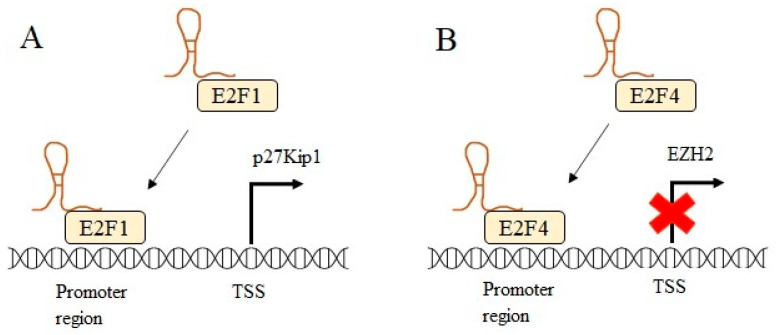
GAS5 acts as a guide by directing transcription factors to the promoter region of the gene. (**A**) GAS5 guides transcriptional activator E2F1 to the promoter region of p27Kip1 and promotes transcription. (**B**) GAS5 enhances transcriptional repression by guiding the transcriptional repressor E2F4 to the promoter region of EZH2. Transcription start site (TSS).

**Figure 5 ncrna-08-00039-f005:**
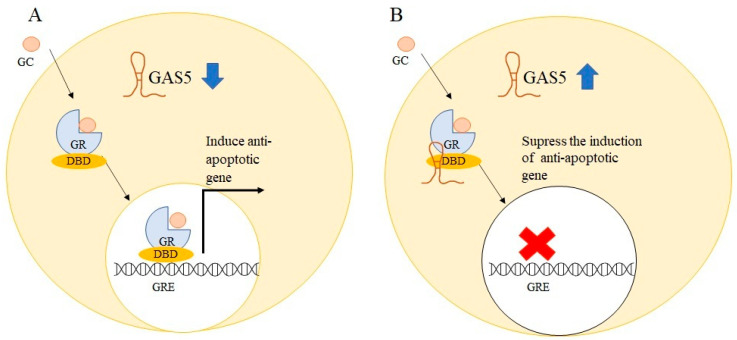
Schematic representation of glucocorticoid receptor (GR) activation by glucocorticoid (GC). (**A**) The glucocorticoid-activated GR bind to the glucocorticoid responsive element (GRE) via the DNA binding domain (DBD) in downregulated GAS5 solid cancer cells to induce anti-apoptotic genes such as *cIAP2*, resulting in cell survival. (**B**) GAS5 acts as a GRE decoy in GAS5 elevated cells, repressing GR-mediated activation of anti-apoptotic genes and therefore sensitising cells to apoptosis.

**Figure 6 ncrna-08-00039-f006:**
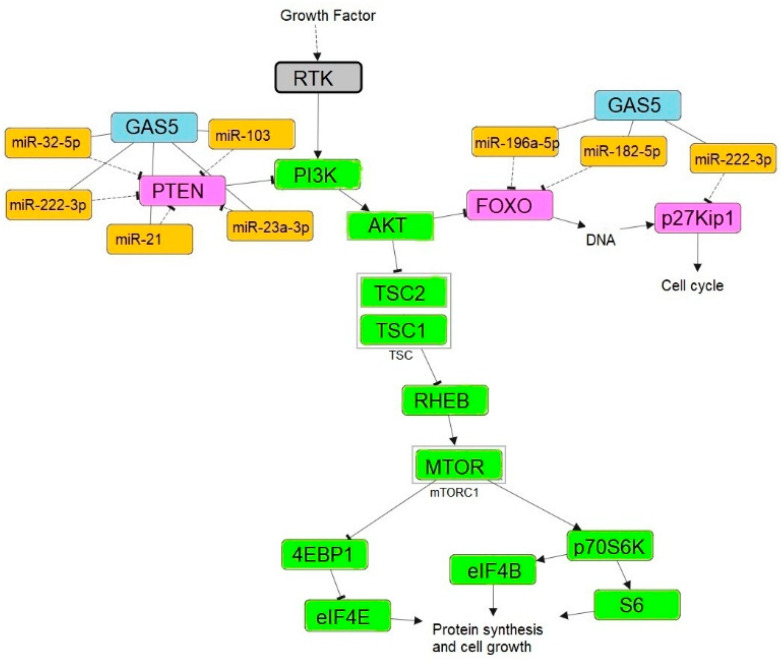
Schematic representation of PI3K/AKT/mTOR pathway. GAS5 inhibits the repression of tumor suppressor gene expression such as *PTEN*, *FOXO* and *p27KIP1* via sponging oncomirs. Pathway was constructed using PathwayMapper [83]. Receptor tyrosine kinase (RTK), Ras homolog enriched in brain (RHEB), Tuberous sclerosis complex 1 and 2 (TSC1 and TSC2).

**Table 1 ncrna-08-00039-t001:** GAS5 regulates PTEN expression via acting as a molecular sponge.

Cancer	GAS5 Expression	miRNA Expression	Mechanism of Action	Reference
Liver cancer	Downregulated in HepG2 and Hep3B cell lines	Overexpression of miR-21 downregulates PTEN expression in HepG2 cells	GAS5 inhibited the miR-21 expression and increased PTEN expression	[45]
Colorectal cancer	Downregulated in HCT116 and SW480 cell lines	miR-222-3p was upregulated in HCT116 and SW480 cell lines compared to NCM460	GAS5 upregulate PTEN via sponging miR-222-3p	[46]
Papillary thyroid carcinoma	Downregulated in BHP5-16, TPC, K1, and BHP2-7 cell lines	miR-222-3p is upregulated in BHP5-16 and K1 cell lines compared to Nthy-ori 3-1	GAS5 upregulates PTEN via sponging miR-222-3p repressing cell proliferation	[47]
Endometrial cancer	Downregulated in HHUA and JEC cell lines	miR-103 is upregulated in endometrial tissues	GAS5 upregulate PTEN by inhibiting miR-103 in endometrial cancer cells	[48]
Pancreatic cancer	Downregulated in PANC-1 and BxPC-3 cell lines	miR-32-5p is upregulated in PANC-1 and BxPC-3 cell lines compared to HPDE6-C7	GAS5 positively regulates PTEN through sponging miR-32-5p	[49]

**Table 2 ncrna-08-00039-t002:** GAS5 polymorphisms in various cancers. ‘-’ indicates that Gas5 expression is not available.

Cancer	Polymorphism	GAS5 Expression	Result	Reference
Acute myeloid leukemia	rs55829688 CC genotype	Elevated	Poor prognosis	[69]
HCC	rs145204276 with Del/Del genotype	Elevated	Increased risk of HCC	[70]
Oral cancer	rs145204276 Ins/Del or Del/Del genotype	Elevated	Poor cell differentiation of oral cancer, advanced tumor stage, and larger tumor size	[25]
Uterine cervical cancer	rs145204276 Ins/Del and Del/Del genotypes	-	low-5 year survival hazard ratio	[71]
Breast cancer	rs145204276 Ins/Del or Del/Del genotype	-	Decreased risk of breast cancer	[73]
Prostate cancer	rs145204276 Ins/Del or Del/Del genotype	-	Reduced risk of lymph node metastasis and seminal vesicle invasion	[74]
Osteosarcoma	rs145204276 Del/Del genotype	Elevated	Decreased risk of osteosarcoma	[27]
Cervical squamous cell carcinoma (CSCC)	rs145204276 Del/Del genotype	Decreased	Increased risk of CSCC	[26]
Urothelial cell carcinoma	rs145204276 Ins/Del or Del/Del genotype	Decreased	Associated with larger tumor size	[77]
Glioma	rs145204276 Ins/Del or Del/Del genotype	Decreased	Increased glioma risk	[72]
Bladder cancer	T G haplotype (rs2067079 and rs6790)	-	Increased risk of bladder cancer	[75]
Nasopharyngeal carcinoma	rs2067079 and rs6790	-	Severe myelosuppression and severe neutropenia	[76]

**Table 3 ncrna-08-00039-t003:** GAS5 regulates chemosensitivity and radiosensitivity in cancer cells.

Cancer	GAS5 Expression in Tissues/Cells	miRNA Expression in Tissues/Cells	Mechanism of Action	Reference
Cervical cancer	Downregulated in radio-resistant tumor tissues and SiHa cell line compared to radiosensitive tissues and ME180 cell line, respectively	miR-106b is upregulated in radioresistant tumor tissues	Overexpression GAS5 upregulates *IER3* via sponging miR-106b	[94]
Esophageal squamous cell carcinoma	Downregulated in ESCC radiation-resistant tissues and radiation-resistant TE-1-R cell line compared to radiosensitive tissues and parent TE-1 cell line, respectively	miR-21 is upregulated in ESCC radiation-resistant tissues and radiation-resistant TE-1-R cell line compared to radiosensitive tissues and arent TE-1 cell line	GAS5 upregulates *RECK* expression via sponging miR-21	[95]
Breast cancer	Downregulated in chemo-resistant breast cancer tissues and chemo-resistant MCF-7 cells compared chemo-sensitive breast tissues and MCF-7 chemo-sensitive cell line, respectively	miR-221-3p is upregulated in chemo-resistant breast cancer tissues and chemo-resistant MCF-7 cells compared to chemo-sensitive breast tissues and MCF-7 chemo-sensitive cell line, respectively,	GAS5 reverses the ABCB1-mediated adriamycin resistance via sponging miR-221-3p and upregulating *DKK2* expression	[93]
Non-small cell lung cancer	Downregulated in cisplatin-resistance A549 and H1299 cell lines compared to parental cell lines	miR-217 is upregulated in negative control cisplatin-resistance A549 and H1299 cell lines compared to GAS5-overexpressed cell lines	GAS5 upregulates *LHPP* expression via sponging miR-217	[96]
Pancreatic cancer	Downregulated in PANC-1, AsPC-1, Capan-2, SW19990, and BxPC3 pancreatic cell lines compared to normal pancreatic epithelial cells, HPDE6-C7	miR-221 is upregulated in PANC-1, AsPC-1, Capan-2, SW19990, and BxPC3 pancreatic cell lines compared to normal pancreatic epithelial cells, HPDE6-C7	Overexpresssion if GAS5 upregulates *SOCS3* expression via sponging miR-221	[97]

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
