# Peer review of "Tumor Suppressive Effects of GAS5 in Cancer Cells"

_ncrna, 2022, doi:10.3390/ncrna8030039_

Round 1

Reviewer 1 Report

In this article, the authors have reviewed the functions of the long non-coding RNA GAS5 in signalling pathways and its roles as a tumor suppressor. Given the importance of this lncRNA in cellular physiology and cancer, this review is timely and useful.

I do feel that this review article would be improved by providing some more molecular insights into GAS5 mode of action. A figure depicting GAS5 structure, highlighting its different important structural elements would be very useful. This figure could show which parts of GAS5 bind miRNAs, interact with eIF4E or the DNA-binding domain of glucocorticoid receptor (GR).

GAS5 is probably packaged into a ribonucleoprotein particle. The authors could provide some information on this issue and how GAS5 binding proteins may regulate its functions and notably its ability to sponge miRNAs.

The authors indicate that GAS5 is translated. Is the role of GAS5 translation known (apart from regulating GAS5 levels through the NMD pathway)? Are small peptides synthesized that could play regulatory functions ?

The authors indicate that GAS5 enhances the binding of transcription factor E2F1 to the p27kip2 promoter (line 140). This is very intriguing and interesting. Is it known how GAS5 could perform such a function ?

Lines 151-153: The message is not entirely clear. One guesses that GAS5 helps recruit the transcription factor E2F4 to the E2H2 promoter and that E2F4 in this context REPRESSES  E2H2 transcription. Is this the correct meaning of the text ? If so, the authors should state explicitly that in this instance, E2F4 acts as a transcriptional repressor.

Minor points:

  • In the abstract, the sentence (lines 16 and 17) “GAS5 is a lncRNA that is highly expressed during the cell growth phase but is downregulated in actively growing cells” is not clear.
  • It is very difficult to extract a “take home message” from the paragraph encompassing lines 204 to 219.
  • Lines 73-74: rephrase: The p706K gene cannot phosphorylate ribosomal protein S6, it is its gene product.
  • Line 200, replace “contrary” by “In contrast”.
  • Line 236: Correct to “MiRNAs are small non-coding RNAs involved in various gene regulations…”
  • Line 275: Correct to “…in resistant cancer cells”.

Author Response

Dear Reviewer 1

First and foremost, we would like to express our gratitude for your helpful feedback, suggestions, and corrections to our review paper. Below are the highlights about the changes that have been made to this manuscript.

1. Reviewer suggested providing more on molecular insights into GAS5 mode of action, by creating figures depicting GAS5, GAS5 roles as molecular sponge, guide, and molecular decoy for glucocorticoid receptor (GR).

We have created few figures as suggested by reviewer:

  • Figure 1, line 70 to depict GAS5 structure and GAS5 transcripts 
  • Figure 3, line 166 which visualizes the GAS5 functions as guide 
  • Figure 4, line 198 which shows GAS5 acting as glucocorticoid receptor element (GRE) decoy 

2. Reviewer also suggested providing some information on GAS5 binding proteins. 

  • We added information on how spliced GAS5 accumulates into messenger   ribonucleoprotein (mRNP) particles, during the growth arrest phase (line 41). 
  • As for GAS5 binding proteins, GAS5 may act as scaffold for protein complexes. However, due to the lack of literature and studies in this area, we are unable to provide more information regarding this matter.

3. Reviewer indicates whether small peptides synthesized during GAS5 translation could play regulatory functions.

Unfortunately, there is a lack of study and experimental data that is able to describe the regulatory role of GAS5 lncRNA-encoded small peptides in cancer. 

4. Reviewer suggested to explain how GAS5 enhances the binding of transcription factor E2F1 to the p27kip2 promoter in the form of a figure. Reviewer also suggests stating E2F4 as a transcriptional repressor.

  • Figure 3 in line 166 explained the GAS5 function as a molecular guide. 
  • The role of E2F4 as transcriptional repressor stated in line 158. GAS5 could reverse the effect of EZH2 in melanoma cells by recruiting transcription factor E2F4, which acts as   transcriptional repressor to the EZH2 promoter region, affecting the EZH2   expression from the transcriptional level and eventually leads to upregulation of tumor suppressor gene CDKN1C.

5. Minor corrections made accordingly as below: 

  • “GAS5 is a lncRNA that is highly expressed during the cell growth phase but is downregulated in actively growing cells” amended to “GAS5 is a lncRNA that is highly expressed during the cell arrest phase but is downregulated in actively growing cells” (line 16-17).
  • Rephrased the whole paragraph explaining GAS5 genetic variants in various cancers (please refer lines 230-247)
  • Amended p706K gene to p706 kinase and also amended “MiRNAs are small non-coding RNAs involved in various gene regulations…” (lines 266-267)
  • Replaced “Contrary” by “In contrast” (lines 224-225)
  • Corrected “…in resistant cancer cells” (line 315) 

Reviewer 2 Report

In this review, the authors focus on a classical—yet still enigmatic—long non-coding RNA, GAS5, specifically highlighting its proposed role in the context of cancer, where it displays tumor-suppressive effects. The authors provide a good introduction to the topic, and cite a wide array of papers, most of which associate GAS5 to specific miRNAs and oncogenes in a way that is not always clear. Unfortunately, part of this is indeed due to the underlying literature.

The mTOR pathway has been tentatively linked to a very large number of miRNAs, but unfortunately most studies do not cross-check each other in the field of “miRNA sponging”, resulting in some of the claims in paragraph 3 about specific miRNAs being tentative and mostly not-yet reproduced.

A similar picture is painted by oncogenes: GAS5 is proposed to lead to growth arrest by (i) sponging oncomirs (ii) upregulating PTEN (iii) blocking c-myc (iv) regulating CDK inhibitors, including p21 and p27, (v) inhibition of EZH2 (vi) reducing cIAP2 activity (vii) upregulating FOXOs. While each (and all) is possible, the overall impression stays quite unclear by simply listing previous studies conducted on GAS5. This type of literature complexity requires an additional layer of analysis to help as a review. For example:

- what is the size (nt) of GAS5?

- where on the GAS5 transcripts are the sponged miRNAs predicted to bind?

- are these “sponging” regions clustered along the transcript?

- are these “sponging” regions conserved across mammalian evolution?

- are these regions adjacent to the briefly-mentioned reading-frame controlling the transcript level via NMD?

In the case of polymorphisms, Table 2 and paragraph 6 give an interesting collection of variants. Interestingly, the lack of correlation between increased/decreased GAS5 levels and the prognosis effect, which should indicate that some of these variants could be separation-of-function alleles. If that is the case, they should occur in more conserved regions of the GAS5 transcript—is this the case? In general, this should be discussed a bit more in the review, here again to move beyond the stage of compiling previous literature.

Figure 1 could gain from representing at least partially the translation mechanism underlying GAS5 degradation. In particular, the short reading frame of GAS5 should be explained/illustrated.

Minor comments:

line 53: correlation FIGO stage/GAS5 expression but it would be clearer to specify that it is a negative correlation (i.e. lower expression, worse diagnosis – and agreeing with the next cited studies in the paragraph).

paragraph 2 (line 68) says that mTOR regulates GAS5 but then paragraph 3.1 (line 82-88) and in fact the end of paragraph 2 too (line 79-80) say that GAS5 regulates mTOR. This feedback loop is interesting and should be clearly stated.

line 148-156: “E2H2” should be “EZH2”

Author Response

Dear Reviewer 2

First and foremost, we would like to express our gratitude for the helpful feedback, suggestions, and corrections to the review paper. Here, we would like to highlight the changes that have been made to this review paper.

1. Reviewer has provided insightful suggestions on improving the review by suggesting additional layers of analysis.

  • We have included the GAS5 transcript diagram in Figure 1. While the sponging regions and conserved regions of miRNA are an interesting area of study, it was not part of our literature search scope for this review.    

2. Reviewer mentioned lack of correlation between increased/decreased GAS5 levels and the prognosis effect in paragraph 6.

  • We highlighted the structural change in  GAS5 variation that could disrupt GAS5 function as a tumor suppressor and SNPs that could change transcriptional activity of GAS5. The lack of correlation of GAS5 expression and prognosis effect among different cancers may be due to the distruption in GAS5 functions as a tumor suppressor in cell specific-manner as mentioned in line 244. 

3. Minor comments

  • We have changed “significant correlation” to “negative correlation” (line 56).
  • We have included Figure 5 as schematic representation of the PI3K/AKT/mTOR pathway to describe the GAS5 role as molecular sponge (line 303).
  • We have corrected “E2H2” to “EZH2” (lines 154 to 162).

Reviewer 3 Report

The authors address a relevant lncRNA with implication for cancer biology. The work is well done. My major concern is the repetitive use of "GAS5 translation'. This is quite inappropriate for a noncoding RNA: please explain and correct throughout the manuscript. 

Author Response

Dear Reviewer 3

First and foremost, we would like to express our gratitude for the helpful feedback, suggestions, and corrections to the review paper. Here, we would like to highlight the changes that have been made to this review paper.

1. Reviewer has mentioned the repetitive use of “GAS5 translation” and inappropriate use for a non-coding RNA.

  • GAS5 translation was only repeated 3 times in this paper. To address this issue, we would like to clarify that GAS5 does undergo translation during the active mTOR signalling pathway. However due to its short reading frame, it is subjected to rapid degradation via NMD pathway which will result in lower GAS5 expression. Schweingruber et al. described that NMD pathway acts as a surveillance pathway that will remove aberrant mRNA transcripts containing premature stop codons during the translation process. GAS5 has premature codon in early exon, hence it will be subjected to degradation via active mTOR signaling pathway.  

Schweingruber, C., Rufener, S. C., Zünd, D., Yamashita, A., & Mühlemann, O. (2013). Nonsense-mediated mRNA decay - mechanisms of substrate mRNA recognition and degradation in mammalian cells. Biochimica et biophysica acta, 1829(6-7), 612–623.

Reviewer 4 Report

The paper is interesting for the field of non-coding RNAs and the information is consistent and well structured. I recommend to make an additional figure including the miRNAs that regulates the cell signaling PI3K/AKT-mTOR sponged by GAS5.

Author Response

Dear Reviewer 4

First and foremost, we would like to express our gratitude for the helpful feedback, suggestions, and corrections to the review paper. Here, we would like to highlight the changes that have been made to this review paper.

1. Reviewer has recommended adding PI3K/AKT-mTOR signaling pathway sponged by GAS5.

  • We have added additional figure including the miRNAs that regulates the cell signaling PI3K/AKT-mTOR sponged by GAS5 in Figure 5.

Round 2

Reviewer 2 Report

In the revised manuscript, the authors have improved the clarity of the text, most notably by the introduction of several figures. This makes the review stronger, and clearer. There is still a difficulty in understanding GAS5 due to the multitude of pathways it has been proposed to participate in. Given the fact that this ncRNA is fairly short, and the large amount of “sponging” miRs cited and participating in its regulation, it would still be important at least to identify which regions of the GAS5 transcript the cited miRNAs bind. This can be added to Fig 1B.

Author Response

Response to Reviewer 2 report (Round 2)

First and foremost, we would like to express our gratitude for your constructive feedback and suggestions. Below are the highlights about the changes that have been made to this manuscript.

Point 1:

Reviewer 2 mentioned that there is still a difficulty in understanding GAS5 due to the multitude of pathways it has been proposed to participate in. Given the fact that this ncRNA is fairly short, and the large amount of “sponging” miRs cited and participating in its regulation, it would still be important at least to identify which regions of the GAS5 transcript the cited miRNAs bind. This can be added to Fig 1B.

Response to point 1:

  • We have added the regions where miRNA binds to GAS5 to Figure 1. This reflects the miRNA binding regions on the GAS5 transcript (linear structure) (line 71 to 77). 
  • However, the detailed miRNA sponging regions on GAS5 secondary structure (variant 1) is shown using force directed graph layout in Figure 2B (line 80 to 83).

We do hope that the reviewer is satisfied with the justification given. 

Thank you.